# Unlocking the Therapeutic Potential of Algae-Derived Compounds in Hematological Malignancies

**DOI:** 10.3390/cancers17020318

**Published:** 2025-01-20

**Authors:** Tamara Vujović, Tina Paradžik, Sanja Babić Brčić, Roberto Piva

**Affiliations:** 1Laboratory for Aquaculture Biotechnology, Division of Materials Chemistry, Rudjer Boskovic Institute, 10000 Zagreb, Croatia; tvujovic@irb.com (T.V.); sanja.babic@irb.hr (S.B.B.); 2Department of Physical Chemistry, Rudjer Boskovic Insitute, 10000 Zagreb, Croatia; tina.paradzik@irb.hr; 3Department of Molecular Biotechnology and Health Sciences, University of Turin, 10126 Turin, Italy; 4Medical Genetics Unit, Città della Salute e della Scienza University Hospital, 10126 Turin, Italy

**Keywords:** algae, antitumor, natural compounds, leukemia, lymphoma, multiple myeloma

## Abstract

Algae could be used as a source of compounds with biological potential as effective antitumor agents. This is particularly relevant for the treatment of hematologic malignancies, which remain a major public health concern due to their high incidence and recurrence rates, despite recent advances in the field. The present review aims to compile evidence on the antitumor properties of algae, focusing on their potential in treating hematologic malignancies. However, the limited isolation and chemical characterization of algal compounds, alongside the scarcity of in vivo studies and clinical trials, represent a major bottleneck in developing algal-based therapies for malignancies. Here, we summarize the currently available findings from various in vitro, in vivo, and clinical studies, highlighting both the opportunities and challenges in harnessing algae-derived compounds for therapeutic use in hematological malignancies.

## 1. Introduction

Algae are a diverse group of aquatic organisms classified based on size, morphology, and structural complexity into macroalgae (seaweed) and microalgae [1]. Macroalgae and microalgae are taxonomically divided into green (Chlorophyta), red (Rhodophyta), and brown algae (Phaeophyceae), whereas the latter are classified into four additional groups: blue-green algae (Cyanobacteriota), yellow-green algae (Xanthophyceae), golden algae (Chrysophyceae), and diatoms (Bacillariophyta) [2]. Due to their diversity and complex lifestyle, algae synthesize different metabolites such as pigments, polysaccharides, polyunsaturated fatty acids (PUFAs), polyphenols, and peptides with outstanding biological activities [3]. Many studies conducted to date have shown that algae-derived compounds could play a role in the prevention and treatment of cancer [4]. These include several dozen studies on hematological malignancies, a diverse group of neoplasms that arise primarily from hematopoietic tissues, including the bone marrow and lymphatic system. These malignancies include various types of cancer, generally classified as leukemia, multiple myeloma (MM), non-Hodgkin’s lymphoma (NHL), and Hodgkin’s lymphoma (HL) [5].

Despite their potential for the treatment of hematological malignancies, there are few studies on the effects of algal extracts and compounds using in vivo assays or their verification in clinical studies. Furthermore, brentuximab vedotin, an antibody-conjugated version of algal peptide dolastatin 10, is the only algal-derived drug approved by the Food and Drug Administration (FDA) for the treatment of hematological malignancies, such as relapsed or refractory systemic anaplastic large cell lymphoma and HL [6].

This review highlights the antitumor potential of algal extracts and compounds and summarizes the so-far-conducted in vivo and in vitro studies on their application in the treatment of hematological malignancies. In addition, insight is given into the possible use of algae as targeted drug delivery systems, as well as the synergistic potential of algal compounds in therapies for hematological malignancies.

## 2. Algae Cultivation and Processing Techniques

Macroalgae can be found in offshore areas and are usually harvested locally from natural sources in smaller quantities [6]. Although the biochemical composition of algae is comparable to that of some terrestrial plants and fungi, algae can grow under extreme conditions (e.g., high salinity and low precipitation levels) and have similar growth rates as higher plants [7]. Low cultivation costs and their ability to adapt to extreme environmental variations provide a significant advantage over terrestrial counterparts [8]. Moreover, algal cultivation does not require adding an excessive amount of nutrients, the use of land, or competition with crops [4]. The cultivation of macroalgae in a natural environment is influenced by seasonal variations (e.g., temperature and sunlight availability) and other biotic and abiotic factors, which makes year-round production challenging [6,9]. To reduce the exploitation of natural sources and increase the productivity of macroalgal cultivation processes, several industrial systems have been developed for their sustainable cultivation in photobioreactors [10]. When developing the novel system for macroalgal farming, it is important to define specific biotic and abiotic factors that have a significant impact on macroalgal yield and sustainability of the cultivation process, such as nutrient availability, temperature, light intensity, and salinity, as well as the susceptibility to diseases [11].

Microalgae are mostly photoautotrophic microorganisms, while some species adapted to low-light environments can grow heterotrophically [12]. Photoautotrophic microalgae use photosynthesis as the main metabolic process that enables them to survive by the utilization of water, sunlight, and carbon dioxide, whereas heterotrophic microalgae need external sources of energy, such as acetate, glycerol, or glucose. In addition, some species are mixotrophic and perform both photosynthesis and nutrient uptake from organic sources [13,14].

Cultivation systems for microalgae have been more extensively studied than macroalgal systems. Microalgae can be grown in open systems (e.g., lakes, lagoons, artificial pools, and canals) or closed systems (photobioreactors) [15,16]. If algae cultivation is carried out on a large scale in open systems, the quality of the algal biomass depends on the harvesting time and the growth location [6,17]. However, one of the biggest drawbacks in developing novel sustainable systems for algal cultivation is the high cost of downstream processes such as drying the harvested biomass, extraction, and purification. Efficient algal harvesting requires specialized techniques to avoid microbial contamination and ensure the quality of the final product. The most commonly used harvesting techniques are centrifugation, filtration, flotation, and flocculation, often employed in combinations to improve the process yield. To increase efficiency, challenges such as the small cell size of the microalgae or unforeseen changes in growing conditions can reduce the biomass yield and make the harvesting process even more expensive [18]. After harvesting, the algal biomass is further processed (e.g., dried or frozen) to preserve the bioactivity of the algal components contained in the collected biomass during storage and successful extraction in further processing procedures [6].

## 3. Algae as a Source of Bioactive Compounds

Both macro- and microalgal biomasses contain various bioactive compounds such as carbohydrates, pigments, proteins, lipids, minerals, etc. For an effective extraction of these compounds, a pre-treatment of the collected algal cells is usually required to break them up and improve mass transfer rates (Figure 1).

The most commonly used methods for cell disruption include a variety of physical, chemical, mechanical, and biological techniques, such as homogenization, bead milling, ultrasonic, solvent, and enzymatic treatments [19]. Among these, the extraction process plays a pivotal role in the separation and purification of algal bioactive compounds. Conventional extraction protocols (Soxhlet extraction, hydro-distillation, maceration, infusion, percolation, and hot continuous extraction) are based on the use of different solvents and include the application of several steps greatly affected by different parameters, e.g., temperature, time, pH, solvent extracting power, sample-to-solvent ratio, particle size, and agitation speed [17,20]. Soxhlet extraction, also known as the solid–liquid extraction method, uses solvents such as ethanol, methanol, acetone, chloroform, or aqueous solutions at varying pressures and temperatures [21]. However, conventional extraction techniques have limitations such as high energy cost, low efficiency, and the extensive use of organic solvents [22].

Innovative extraction methods are, therefore, currently being developed that are more efficient, economical, and environmentally friendly. These include pressurized fluid/liquid extraction (PLE), subcritical water extraction (SWE), microwave-assisted extraction (MAE), enzyme-assisted extraction (EAE), ultrasound-assisted extraction (UAE), and supercritical fluid extraction (SFE) [21,23]. Among these, PLE and SWE are most commonly used for algal matrixes. PLE uses high-temperature (max. 200 °C) and high-pressure (3.5–20 MPa) solvents such as water, ethanol, or their combination to increase the extraction yield of the analyte. Similarly, the SWE method uses water at high temperatures and pressures below its critical point, reducing its dielectric constant to make it comparable to commonly used organic solvents such as methanol or ethanol [20,21].

These novel techniques have many advantages over conventional methods. The most important are shorter extraction time, higher extraction yield, lower solvent and energy consumption, and the better preservation of unstable compounds. However, the optimization of the extraction parameters is crucial to minimize the negative impact on the yield, bioactivity, and chemical structure of the target compounds [17].

## 4. Algae as Therapeutic Agents in Treating Hematological Malignancies

Due to the algal diversity, there are significant differences in their physiochemical and biological properties [22]. Both macro- and microalgal biomasses consist of various compounds such as carbohydrates, pigments, proteins and peptides, lipids, polyphenols, and other compounds with potential antioxidant, anti-inflammatory, antibacterial, antifungal, antiviral, and antitumor effects [24,25]. The anticancer effects of algal compounds (Figure 2) are mostly attributed to their antioxidant and anti-inflammatory activities, resulting in immunostimulation, the promotion of tumor suppressor gene expression, the inhibition of tumor angiogenesis, or the induction of apoptosis [6,26]. Furthermore, these compounds inhibit metastatic ability by modulating key pathways such as PI3K/AKT, JAK/STAT, and HIF-1α, suppressing matrix-degrading enzymes like MMPs, reducing cell adhesion and migration, and disrupting the tumor microenvironment through antioxidant, anti-inflammatory, and anti-angiogenic mechanisms [27,28,29,30].

Hematological malignancies often require complex, multi-modal treatments like chemotherapy, radiation, and stem cell transplants. Recently introduced alternative approaches, such as targeted therapies and immunotherapies, have improved disease outcomes and reduced toxicity [31,32]. The development of modern technologies, such as next-generation sequencing, multiparametric flow cytometry, and molecular genetics, has enabled the identification of specific genetic mutations and biomarkers that can be used to classify and diagnose hematologic malignancies [33,34].

Although survival rates for patients with hematologic malignancies have improved dramatically over the past decade, these diseases remain a major public health problem worldwide with increasing impact in many countries [35].

Leukemia develops in the bone marrow and is usually associated with the rapid production of abnormal white blood cells that outgrow healthy cells, impairing immune function. It is the most common type of hematological malignancy and involves several subtypes, depending on the origin (myeloid or lymphoid), the growth rate (acute or chronic), and genetic abnormalities [35,36]. Targeted agents, such as tyrosine kinase inhibitors and immunotherapies, have shown improved therapeutic efficacy in specific subgroups of patients associated with reduced adverse effects. Furthermore, the development of chimeric antigen receptor T-cell therapy represents a breakthrough in acute lymphoblastic leukemia treatment, resulting in remarkable responses and potential long-term remissions [36,37].

Lymphomas, which arise from lymphocytes in the lymphatic system, include HL and NHL lymphomas and involve abnormal cell proliferation in lymph nodes or other lymphoid tissues [38]. The presence of Reed–Sternberg cells is characteristic of HLs, which account for 10–15% of all lymphoma cases, whereas the other 85% are NHLs that can be further subdivided into T-cell, B-cell, and natural killer/T-cell NHLs [39]. Chemotherapy for lymphomas depends on the specific subtype requiring tailored therapeutic protocols [38]. While patients with HL are treated with a specific chemotherapy regimen in combination with radiotherapy, the most commonly used treatment for NHL is the combination of cyclophosphamide, doxorubicin, vincristine, and prednisone [40,41]. Regardless of the progress made in the development of more efficient therapies for the treatment of patients with lymphomas, the response of patients with refractory or relapsed lymphomas to the commonly used treatments is often insufficient [42]. However, immunotherapies (e.g., cellular treatments and immune checkpoint inhibitors) have recently emerged as promising alternatives for the treatment of patients with malignant lymphomas. The antitumor activity of this type of therapy is based on blocking the immune evasion of malignant cells and enhancing the activity of immune cells, which represents encouraging progress in lymphoma treatments [43].

MM is a malignancy of terminally differentiated plasma cells and accounts for approximately 10% of hematologic malignancies in developed countries [44]. Because this disease is characteristic of the elderly population, there is an increased incidence of MM worldwide, particularly in high-income countries [45]. MM is characterized by the genetic heterogeneity and overproduction of immunoglobulins and, therefore, depends on the precise regulation of protein degradation systems [46]. As a result, myeloma cells are highly sensitive to proteasome inhibition. Several proteasome inhibitors have been successfully developed, tested, and approved for routine clinical use [47,48]. Despite the success of proteasome inhibitors in MM treatment, resistance and adverse toxic effects such as peripheral neuropathy and cardiotoxicity could arise [46].

### 4.1. In Vitro Studies

#### 4.1.1. Algal Extracts

Algal extracts comprise a vast array of bioactive compounds with potential therapeutic applications. Since the extraction process results in a higher yield than the fractionation and isolation of specific compounds, extracts are often used to investigate the bioactive potential of algae, including their antitumor properties.

The antiproliferative activity of green macroalga *Caulerpa lentillifera* was tested [49]. This alga is produced in aquaculture and is traditionally consumed in the cuisines of the Pacific and Southeast Asian regions. Different *C. lentillifera* extracts showed low cytotoxicity against normal skin fibroblasts, epithelial cells, and peripheral blood mononuclear cells (PBMCs), suggesting their safety and potential as antioxidants and anti-obesity agents. Importantly, *C. lentillifera* extracts were toxic to several cancer cell lines, including the acute myeloid leukemia cell line HL-60 (ethanol macerate, inhibitory concentration (IC_50_) = 315 µg/mL).

Acetone extracts from Phaeophyceae, Rhodophyceae, and Chlorophyceae species collected in Mexico—specifically *Rhizoclonium riparium*, *Spyridia filamentosa*, and *Caulerpa sertularioides*—exhibited significant cytotoxic effects in murine B-cell lymphoma cells [50]. These effects included reduced cell proliferation and the induction of apoptosis. Notably, the acetone extracts of *S. filamentosa* and *R. riparium* significantly suppressed proliferation at all tested concentrations (12.5–100 µg/mL), while *C. sertularioides* showed the most pronounced antiproliferative effect, reducing cell viability to less than 40% at 100 µg/mL. In contrast, methanol and hexane extracts from the same macroalgal species showed negligible effects on cell proliferation. The acetone extracts consistently displayed the highest antioxidant, antimutagenic, and antiproliferative activities. These results were attributed to the high content of flavonoids and chlorophyll, with phenolic compounds and carotenoids contributing to a lesser extent. These findings highlight the potential of acetone extracts from macroalgae as a rich source of bioactive compounds with potential therapeutic applications.

In addition to macroalgae, in vitro studies have focused on the investigation of anticancer properties of various microalgal species [51]. Hexane extracts of *Chlorella* sp. isolated from the desert in Qatar showed high antiproliferative activity against leukemia cells (IC_50_ < 100 µg/mL). This species is intriguing because it can tolerate high temperatures, up to 40 °C, making it suitable for large-scale pond production. In addition, *Chlorella* is rich in essential fatty acids and carotenoids, which are suitable for nutraceutical applications [52].

Methanol extracts derived from lipid-enriched microalgae species (*Stichococcus bacillaris*, *Phaeodactylum tricornutum*, *Microcystis aeruginosa*, and *Nannochloropsis oculata*) showed significant antileukemic potential. Essential oil extracts from these microalgae were analyzed using GC-MS, leading to the identification of twelve bioactive molecules potentially responsible for their cytotoxic effects [53]. In silico docking analysis was used to predict the molecular targets of five essential oil constituents. The results suggested that 9-octadecenoic acid, methyl ester, (9Z)-targets Akt and Bcl-xL; 2,2,4-trimethyl-1,3-pentanediyl bis(2-methylpropanoate) interacts with caspase-3; 5,6-dihydroergosterol targets Mdm-2; tricosane is associated with Mek-1 and PARP-1; and 9-octadecenamide (Z)- is predicted to target p38. Several in vitro studies have reported on the effect of the extracts and compounds isolated from algae by increasing or decreasing oxidative stress. For example, an extract of *Skeletonema marinoi* induced apoptosis in K562 cells after 48 h at 0.75 mg/mL, while it did not affect Vero cells derived from kidney [54]. A significant increase in malondialdehyde production was measured in untreated K562 cells, compared to Vero, indicating oxidative stress. These cells also showed a low activity of enzymes involved in the oxidative stress response. Treatment with the microalgae resulted in a significant recovery of SOD, CAT, and GPx levels compared to untreated K562 cells [54]. An excessive production of reactive oxygen species (ROS) can promote tumor cell survival, proliferation, and drug resistance [55]. The *S. marinoi* extract reduced oxidative stress through the NOX2 pathway and decreased NO levels in K562 cells, which is significant as NO production is known to cause nitrosative stress and consequently reduce the effect of anticancer drugs [56]. In another study, ethyl acetate extracts of the marine alga *Colpomenia sinuosa* induced apoptosis in HL-60 and U937 leukemia cells by ROS formation, which was also prevented by the addition of N-acetylcysteine (NAC) [57].

Unfortunately, other studies on the antiproliferative effects of extracts from algal species included little or no data on toxicity to normal cell lines or in vivo data. However, a lot of preliminary data found in the literature may be useful for future directions. In Table 1, we presented a summary of studies investigating the antitumor properties of algal extracts that could potentially be used in treating hematological malignancies.

#### 4.1.2. Algal Compounds

##### Pigments

Beyond their well-known role in capturing light energy for photosynthesis, algal pigments have gained attention for their therapeutic properties, particularly in the context of hematological malignancies. Numerous pigments showed significant antiproliferative activity, primarily inducing apoptosis, arresting the cell cycle, and modulating oxidative stress. The ability of algal pigments to selectively target cancer cells makes them attractive for further research as possible potential therapeutics for solid tumors and hematological malignancies [61]. Algal pigments can be classified into three main groups: chlorophylls, carotenoids, and phycobiliproteins, with carotenoids showing the greatest anticancer potential [25,62,63].

Among the carotenoids, astaxanthin and fucoxanthin have the greatest anticancer potential in vitro and in vivo [6]. Astaxanthin has shown promising activity in colorectal, breast, and hepatocarcinoma models, as well as in melanoma, by inhibiting cell cycle progression or inducing apoptosis [64]. Currently, the most abundant source of astaxanthin is the microalga *Haematococcus pluvialis* [65]. β-Carotene isolated from several microalgae, such as *Dunaliella salina*, *Chlorella vulgaris*, *Spirulina* sp., and *Asterarcys quadricellulare*, has been reported to induce the apoptosis of prostate cancer cells and could potentially be used to suppress colon and lung cancer cells [64,66]. Moreover, lycopene isolated from green microalga *Chlorella marina* induced antiproliferation and caused apoptosis in human prostate cancer [67].

β-Cryptoxanthin, astaxanthin, and β-carotene reduce leukemia cell proliferation and cause apoptosis while showing low toxicity to normal cells. Mechanistically, astaxanthin and β-carotene increase pro-apoptotic signaling, reduce telomerase activity, and alter NF-B, while β-cryptoxanthin causes apoptosis and cell cycle arrest through PPARγ and p21. Their unique and distinct mechanisms, including the effect of β-cryptoxanthin on cell cycle regulation and astaxanthin and β-carotene’s suppression of telomerase, highlight their complementary therapeutic potential [61].

Fucoxanthin isolated from several algal species (e.g., brown macroalgae *Undaria pinnatifida*, *Fucus* spp., *Sargassum* spp., *Laminaria religiosa*, *Petalonia binghamiae*, *Scytosiphon lomentaria*, and microalga *Phaeodactylum tricornutum*) inhibits colon cancer cells and neuroblastoma [20,25]. In addition, this pigment showed antimetastatic properties in several cancer types, through the inhibition of PI3K/Akt/mTOR and the downstream protein matrix metalloproteinase MMP-2 and MMP-9 [68]. Also, it was found that it inhibits tumor-induced lymphangiogenesis in vitro and in vivo [69].

Overall, fucoxanthins from algae have been intensively studied for their effect on the leukemia cell lines.

Fucoxanthin and its metabolite fucoxanthinol, found in brown macroalga *Undaria pinnatifida*, showed significant anti-cancer activity by targeting adult T-cell leukemia (ATL) induced by the human T-cell leukemia virus type 1 (HTLV-1). These compounds exhibited potent antileukemic effects by inducing apoptosis through caspase activation and downregulating antiapoptotic proteins like B-cell lymphoma 2 (Bcl-2) and survivin. Notably, natural fucoxanthin and fucoxanthinol were not toxic to uninfected cell lines and PBMCs, confirming their selective cytotoxicity towards cancer cells. In a mouse model, the oral administration of fucoxanthinol significantly reduced tumor growth and increased apoptotic markers in tumors without toxicity or other side effects in vivo [70].

Yamamoto et al. [71] investigated the therapeutic potential of fucoxanthin and fucoxanthinol, extracted from brown macroalga *Cladosiphon okamuranus*, in the treatment of primary effusion lymphoma (PEL), a rare and aggressive NHL associated with human herpes virus 8 (HHV-8). Both compounds significantly reduced PEL cell viability in a dose-dependent manner, with fucoxanthinol showing greater potency. They induced apoptosis via a caspase-dependent mechanism and caused G1 cell cycle arrest by downregulating key regulators such as cyclin D2 and CDK4. In addition, fucoxanthin and fucoxanthinol targeted key oncogenic pathways, including NF-B, AP-1, and PI3K/Akt, leading to the decreased expression of anti-apoptotic proteins such as Bcl-xL and survivin. Moreover, fucoxanthin was shown to induce apoptosis in HL-60 cells through increased oxidative stress. Fucoxanthin isolated from the brown macroalga *Ishige okamurae* induced the apoptosis of HL-60 cells through Bcl-xL signaling pathways, mediated by the generation of ROS [72].

Siphonaxanthin, a well-known carotenoid with structural and functional similarities to fucoxanthin, has unique variations that increase its cytotoxicity on HL-60 leukemia cells [73]. For example, siphonaxanthin significantly decreased cell viability within 6 h of treatment with 20 μM, while fucoxanthin required longer exposure time to achieve comparable results. This rapid and more pronounced cytotoxic activity is probably due to the hydroxyl group, which improves cellular uptake and amplifies its biological activity. Further research by Almeida et al. [74] has shown that siphonaxanthin induces cell cycle arrest and sensitizes leukemia cells to TRAIL-mediated apoptosis by upregulating death receptors.

Phycobiliproteins are water-soluble pigments that can be further divided into three classes: phycoerythrins (red), allophycocyanins (light blue), and phycocyanins (blue) [75]. Phycobiliproteins detected in many algal species, mainly cyanobacteria *Arthorspira* sp. and red macroalgae *Porphyra* sp., *Pyropia tenera*, and *Porphyridium cruentum*, also show anticancer activity in lung (A549) and liver (HepG2) cancer cells, exhibiting immunomodulatory, anti-inflammatory, and antioxidant effects [6,19,25].

R-phycoerythrin, a pigment extracted from the red alga *Solieria filiformis*, has shown the ability to induce apoptosis in leukemia cells through the activation of caspase pathways and DNA breakage. It reduces ROS by disrupting the oxidative balance necessary for cancer cell survival and arrests cell proliferation at the G0/G1 phase of the cell cycle. Its selective cytotoxicity specifically targets leukemia cells while sparing normal ones, making it a promising alternative to traditional treatments with fewer side effects [76].

Scytonemin, a photoprotective pigment isolated from cyanobacteria, has been identified as a strong inhibitor of polo-like kinase 1 (PLK1), an important cell cycle regulator. Many malignancies are characterized by high PLK1 levels, which are associated with aggressive disease progression and worse prognosis. The study of Zhang et al. [77] showed that scytonemin effectively suppresses cell proliferation in three MM cell lines (U266, RPMI8226, and NCI-H929) and causes cell cycle arrest in the G2-M phase. This mechanism is achieved by downregulating PLK1 activity without changing its expression. The ability of scytonemin to induce cell cycle arrest and stop the growth of tumor cells makes it a promising therapeutic agent for MM, particularly for patients who have developed resistance to conventional treatments. To date, several studies have reported on the potential of microalgal pigments for the treatment of hematological malignancies, but none of them have used microalgal biomass to extract bioactive pigments. Therefore, studies investigating the potential application of pigments directly isolated from algae in treatments of hematological malignancies are still limited to a few macroalgal species. Table 2 summarizes the macroalgal pigments that have been studied with different cell lines characteristic of hematological malignancies.

##### Polysaccharides

Polysaccharides are polymers composed of various monosaccharides such as D-xylose, D-fructose, D-glucose, D-mannose, D- and L-galactose, and L-arabinose linked by glycosidic bonds [78]. The main functions of polysaccharides in algal cells are energy storage (e.g., carrageenan, glucan, agar, or fucoidan) and structural components (e.g., lignin, hemicellulose, and cellulose) [25].

The antitumor activity of algae-derived polysaccharides is mostly attributed to the presence of the sulfate group [79]. Several studies have reported that sulfated polysaccharides derived from green macroalgae *Caulerpa prolifera*, *Caulerpa racemosa*, and different *Ulva* induce apoptosis in various cancer cell lines. Ulvan, a sulfated polysaccharide extracted from different *Ulva* species inhibits the proliferation of hepatocellular carcinoma tumor cells and also exhibits anticancer activity in breast, cervical, and colon cancer in vitro [80,81,82]. Carrageenan, another sulfated polysaccharide isolated from several red algae (e.g., *Jania rubens*, *Gracilaria caudata*, and *Gelidium amansii*), could also be used as an anticancer agent [79].

Laminarins, alginates, and fucoidans are polysaccharides found only in brown macroalgae, accounting for over 50% of their dry weight [17,43]. They contain large amounts of cell wall polysaccharides, many of which are fucose-containing sulfated polysaccharides, including fucoidans [83,84]. The composition and bioactivity of fucoidans can vary depending on the species of algae. Fucoidans have been shown to exhibit a wide range of bioactivities, including the inhibition of tumor growth through the induction of apoptosis and cell cycle arrest, the inhibition of metastasis, and the enhancement of the toxic effects of chemotherapeutic agents [84,85]. Fucoidan from brown macroalgae is found to suppress the epithelial-to-mesenchymal transition by the downregulation of the HIF1-α signaling pathway in triple-negative breast cancer cell lines [86]. Furthermore, it was shown that fucoidan can inhibit angiogenesis in MM by the downregulation of the HIF-1α/VEGF proteins’ hypoxia that was possibly associated with the PI3K/AKT pathway [87]. In the context of MM, the chemotherapeutic drug cytarabine can induce cell escape by upregulating CXCR4, facilitating tumor cell adhesion to the bone marrow stroma. Fucoidan has been shown to interfere with this process by disrupting the protective niche that supports tumor survival and resistance to chemotherapy. This is probably achieved via the downregulation of CXCR4, MMP-9, and RhoC [88].

Monla et al. [89] investigated the anticancer properties of two polysaccharides (alginate and fucoidan) isolated from brown macroalga *Colpomenia sinuosa*, showing that the combination of these polysaccharides with vitamin C causes apoptosis of colon cancer cells. In addition, fucoidan from *Undaria pinnatifida* induced apoptosis in the hepatocellular carcinoma cell line by generating ROS [90]. Moreover, several studies investigated their antiproliferative effect against hematologic malignancies. The antiproliferative activity of the fucoidan-containing polysaccharide fraction of the brown alga *Sargassum polycystum* was demonstrated against the HL-60 leukemia cell line with an IC_50_ of 84.63 µg/mL, while it showed no effect against epithelial cells. Increasing concentrations of this fraction induced DNA damage, apoptosis, and necrosis in the tested cell lines, including leukemia cells [37].

In another study, polysaccharides were isolated from several species of macroalgae, including *Laminaria ochroleuca*. Polysaccharides with a higher degree of sulfation exhibited the strongest bioactivity [91] and showed antioxidant and immunomodulatory properties, including macrophage activation. The anticancer effects were evaluated across several different cancer cell lines, with the leukemia cell line U937 showing an IC_50_ value of 3.72 mg/mL. Commercial fucoidan was isolated from brown alga *Laminaria japonica* and purified by chitosan microsphere [92]. Such fucoidan had a high degree of sulfation and inhibited the proliferation of the human acute myeloid leukemia cell line THP-1 with a concentration of 200 μg/mL. However, this effect was modest with 60% of cells remaining viable alive cells at 72 h. Interestingly, while fucoidan inhibited angiogenesis, its desulfated form did not, indicating that sulfate groups were required to prevent the process of angiogenesis.

Fucoidan was also isolated from the brown alga *C. okamuranus* and showed activity against ATL, caused by human T-cell leukemia virus type 1 (HTLV-1). This fucoidan inhibited the growth of PBMCs in ATL patients and HTLV-1-infected T-cell lines, while normal PBMCs were spared [70,93]. Further research revealed that fucoidan inhibited PBK/TOPK-AKT signaling in MT-2 cells by modulating the phosphorylation of PBK/TOPK and the downregulation of kinase Akt [70].

Sulfated polysaccharides have been isolated from the green algae *Caulerpa lentillifera* and *Caulerpa racemosa* [49], which also exhibit a broad spectrum of biological effects (antioxidant, antidiabetic, and antiaging). While fucose monosaccharides are mainly found in brown algae, xylose monosaccharides and other types of sulphate (type B-ulvanobiuronic acid-3-sulfate and type A-ulvanobiuronic acid-3-sulfate) were observed in these green algae. The polysaccharides of both types had a good anti-cancer effect in several different cell lines. However, *C. lentillifera* displayed a stronger effect on the leukemia cell line (IC_50_ = 305 µg/mL) with lower cytotoxicity on normal cells (6 mg/mL for PBMCs and 3 mg/mL for human epithelium).

Extracted polysaccharide fractions from *Codium bernabei*, a green alga growing on Chilean coasts, consists mainly of sulfated galactan units [94]. The acidic fraction of polysaccharides showed better antiproliferative effects on leukemia cell line HL-60 than on other tested cells (colon and breast cancer).

Fucose-containing sulfated polysaccharides from *Saccharina latissima* interfered with the CXCL12/CXCR4 axis in human Burkitt’s lymphoma cells with downstream effects like the migration and secretion of matrix metalloproteinase-9 [95].

Microalgal polysaccharides have also been studied as promising candidates for antitumor therapies. Polysaccharides isolated from microalgae are usually used as thickeners or gelling agents in the food industry. However, some studies reported that paramylon (β-1,3-glucan) isolated from microalga *Euglena gracilis* may have antitumor properties [19,25,96].

Exopolysaccharides (EPSs) were extracted from the marine microalga *Tetraselmis suecica* grown in autotrophic or heterotrophic conditions [97]. Heterotrophic growth resulted in significantly better EPS production and increased antioxidant activity compared to autotrophic conditions. Interestingly, the acidic EPS fraction had much better antiproliferative activity against several cell lines than total EPS, including leukemia HL-60 (acidic autotrophic and heterotrophic IC_50_ = 36 µg/mL and 68 µg/mL, respectively), but also remarkable cytotoxicity against gingival fibroblast cell line (IC_50_ = 165 µg/mL and 61 µg/mL, respectively).

An extracellular polysaccharide from the marine microalga dinoflagellate *Gymnodinium* sp. A3, described as a D-galactan sulfate (GA3P) in combination with L(+)-lactic acid, showed cytotoxicity against human lymphoid cells, especially to the T-cell leukemia cell MT-4 [98]. GA3P is a potent inhibitor of DNA topoisomerase (Topo) I and II, irrespective of the presence or absence of the lactate group [99]. In addition, microalga *Gymnodinium* sp. A3 produces exopolysaccharide GA3P (D-galactan sulfate associated with L-(+)-lactic acid), which induces apoptosis in K562 cells, supposedly as a result of an interaction between GAP3 and the cell surface receptors.

*Porphyridium cruentum* is another microalga that can produce a large amount of EPS [100]. This EPS is mainly composed of α-glucan and had moderate immunomodulatory properties in vivo. However, the measured cytotoxic activities against several cell lines were not pronounced, with leukemia U937 exhibiting IC_50_ of 1.6 mg/mL in the presence of EPS.

Overall, the antitumor activity of algal polysaccharides depends on their structure and algal source. However, most of them have a modulating effect on cell proliferation and induce apoptosis in various cancer cells, making them potential natural sources of novel drugs for future therapies against hematological malignancies and other cancers [81,101,102]. Table 3 provides an overview of the in vitro studies conducted with algal polysaccharides and the list of hematologic malignant cell lines.

##### Polyunsaturated Fatty Acids (PUFAs)

Lipids are a group of water-insoluble macromolecules consisting of various compounds with biological potential, such as glycerides, phospholipids, sterols, and fatty acids [65]. Along with alpha-linolenic acid (ALA), docosahexaenoic acid (DHA), and eicosapentaenoic acid (EPA) play an important role in the development of the retina, brain, and reproductive tissues in infants and their function in adults. Microalgae, especially species from genera *Nannochloropsis* and *Schizochytrium*, are one of the main producers of DHA and EPA, long-chain PUFAs that cannot be synthesized by humans [7,25,104]. PUFAs extracted from other microalgae (e.g., *Crypthecodinium*, *Spirulina*, and *Haematococcus*) could be effective as novel therapies for lung, liver, prostate, colorectal, kidney, breast, and ovarian cancer [66,105,106]. For example, Yao et al. [107] reported the inhibition of lung adenocarcinoma cells following treatment with DHA or EPA. Additionally, DHA isolated from microalga *Crypthecodinium cohnii* demonstrated significant antileukemia effects by regulating cell growth and inducing apoptosis through the upregulation of Bax and the inactivation of Rb proteins [108]. DHA from other algal sources, such as the green microalga *Ostreococcus tauri* and diatom *Thalassiosira pseudonana*, could also serve as a source of this bioactive compound [25,109]. Furthermore, Zha et al. [109] isolated the fatty acid (±)-(E)-12-hydroxyoctadec-10-enoic acid (HOEA) from the red macroalga *Tricleocarpa jejuensis*, which exhibited cytotoxic effects against U937 leukemia cells by inducing caspase-3 and caspase-8 activation. Finally, nonyl 8-acetoxy-6-methyloctanoate, a fatty alcohol ester isolated from diatom *Phaeodactylum tricornutum*, induced DNA damage and apoptosis in HL-60 cells through the caspase-3 and p53-mediated pathway [51,110].

##### Peptides

Algae are a rich source of protein, with microalgae such as *Chlorella* and *Spirulina*, containing protein contents of 50–70% of their dry weight [19]. In contrast, macroalgae generally have a protein content of 5–20%; however, certain red algae species such as *Porphyra* spp. and *Palmaria* spp. can reach up to 47% of their dry weight [23,79]. Protein hydrolysis produces peptides that exhibit various anticancer properties through mechanisms such as the inhibition of angiogenesis, the disruption of cell cycle progression, and the induction of apoptosis. For example, peptides isolated from the green microalga *Chlorella vulgaris* can prevent cell damage and have the potential to act as antitumor agents [64,104,111].

Wang and Zhang [112] reported that a polypeptide isolated from green microalga *Chlorella pyrenoidosa* inhibited the growth of human liver cancer cells. Also, peptides derived from the protein hydrolysate of microalga *Navicula incerta* showed antihepatotoxic activities on HepG2/CYP2E1 cells [113].

Wrasidlo et al. [114] demonstrated the antitumor potential of the lipopeptide somocystinamide A isolated from *Lyngbya majuscula* on the Jurkat and CEM cell lines. The cytotoxic activity was attributed to the inhibition of angiogenesis and the induction of apoptosis through caspase 8 activation.

Algal peptides offer several advantages as chemotherapeutic agents over antibodies or proteins. These include low toxicity, easier synthesis, and higher selectivity and specificity [115]. Furthermore, recent studies suggest that the integration of algal peptides into novel therapies could enhance the effectiveness of cancer treatments [111].

Another group of algal compounds with diverse bioactive properties is the ester-bonded class of peptides known as depsipeptides. For example, kahalalides P2 and Q3, cyclic depsipeptides isolated from green macroalga *Bryopsis* sp., showed moderate inhibitory activity in the HL-60 cell line [67]. Aurilides, another class of depsipeptides mainly found in cyanobacteria (e.g., *Lyngbya majuscula*), also showed cytotoxicity against various leukemia cell lines including HL-60, K562, MOLT-4, RPMI-8226, and SR [51,116,117]. In addition, criptophycin 1 isolated from cyanobacterium *Nostoc* sp. displayed cytotoxic activity against L1210 leukemia cells [118]. Apratoxin A, a depsipeptide found in *Lyngbya* sp. and *Symploca* sp., induced apoptosis by blocking the cell cycle in G1 phase via the JAK-STAT pathway [51]. Lagunamides A, B, and C, cyclic depsipeptides derived from *Lyngbya majuscula*, showed strong cytotoxic effects on P388 cells, with IC_50_ values of 6.4, 20.5, and 24.4 nM, respectively [119,120].

Lectins, non-immune proteins that specifically and reversibly bind to carbohydrates, also exhibit antitumor properties. Xu et al. [121] identified a novel lectin called Siye in the genome of the red alga *Kappaphycus alvarezii* and investigated its antitumor properties on four different cell lines, including HL-60. In the human promyelocytic leukemia cell line, Siye induced cytotoxicity within 24 h of treatment by regulating the p53 and caspase pathways.

##### Phenolic Compounds

Phenolic compounds form a large group of natural molecules that are characterized by the presence of one or more hydroxylated aromatic rings in their structure. The best-known phenolic compounds isolated from algae are flavonoids, phenolic terpenoids, bromophenols, and phlorotannins, with the latter being found only in marine macroalgae. Phenolic compounds have a wide range of bioactive properties, including anticancer [7,23].

Bromophenols isolated from red alga *Rhodomela confervoides*, exhibited cytotoxicity against KB (a human carcinoma of the nasopharyngeal cell line) and A549 (human lung adenocarcinoma epithelial cell line) [122]. In addition, several in vitro studies have investigated the effects of this algal compound on oxidative stress. Specifically, some bromophenol derivatives found in marine algae showed protective effects on H_2_O_2_-induced ROS formation in keratinocytes by increasing the expression of the antioxidant proteins TrxR1 and HO-1 [123]. Although some of these compounds induced apoptosis in K562 leukemia cells, the mechanism of action is not known.

Phlorotannins are another class of phenolic compounds consisting of phloroglucinolic (1,3,5-trihydroxybenzene) units that show great potential for use as anticancer agents [124]. For example, dieckol, dioxinodehydroeckol, and phlorofucofuroeckol, phlorotannins derived from macroalga *Ecklonia cava*, expressed antitumor properties against different cancer cell lines [125]. Among brown macroalgae, phlorotannins derived from *Alaria esculenta*, *Ascophyllum nodosum*, *Bifurcaria bifurcata*, *Fucus vesiculosus*, *Laminaria japonica*, and *Sargassum muticum* showed an antiproliferative effect on different cell lines, including leukemia [75].

Other promising sources of phlorotannins are algae from the genera *Eisenia*, *Ishige*, and *Cystoseira* [126]. Phlorotannin-derived compounds can promote the apoptosis of cancer cells by activating caspases [124,127]. Specifically, phlorotannins extracted from brown alga *Laminaria japonica* had anti-proliferative activity on murine leukemia cells (P388) (IC_50_ = 120 μg/mL), with authors suggesting that the observed antitumor properties may be related to the ability of phlorotannins to scavenge free radicals and carcinogenic agents [128]. Finally, a bis-prenylated quinone isolated from the brown macroalga *Perithalia capillaris* inhibited the HL-60 cell proliferation (IC_50_ = 0.34 μM) [129].

Interestingly, phloroglucinol isolated from brown macroalgae showed no cytotoxic activity but suppressed the metastatic ability of breast cancer cells through the downregulation of SLUG by inhibiting PI3K/AKT and RAS/RAF-1/ERK signaling. These results were also confirmed in vivo [130].

Studies investigating the anticancer properties of phenolics are mostly conducted with algal extracts rich in various polyphenolic compounds as the algae contain only small amounts of specific phenolic compounds [126]. Therefore, cultivation and extraction methods should be further developed to obtain larger amounts of phenolic compounds from algal biomasses. Also, studies of toxicity on non-cancerous cell lines, as well as in vivo and clinical studies, should be conducted to investigate the potential of phenolic compounds in the treatment of various malignant diseases.

Table 4 summarizes the algae-derived compounds that show potential for the treatment of hematological malignancies in addition to pigments and polysaccharides.

### 4.2. Synergistic Potential of Algal Compounds in Hematological Malignancy Therapies

The combination of algae-derived natural compounds with conventional treatments has shown better therapeutic results in hematological malignancies. These substances can target different metabolic pathways, efficiently overcome drug resistance, and increase efficacy when used in conjunction with therapies such as chemotherapeutics or tyrosine kinase inhibitors (TKIs).

Algae-derived carotenoids have shown the potential to overcome multidrug resistance (MDR) in cancer cells. A study by Gyémánt et al. [132] highlighted the ability of carotenoids, such as capsanthin and zeaxanthin, to reverse MDR by inhibiting P-glycoprotein also known as multidrug resistance protein 1 (MDR1) or ATP-binding cassette sub-family B member 1 (ABCB1), an efflux pump that reduces intracellular drug concentrations in resistant cancer cells. Notably, these carotenoids have been shown to increase the intracellular retention of epirubicin and significantly induce apoptosis in drug-resistant leukemia and lymphoma cell lines.

Molnár et al. [133] provided further insights into the function of carotenoids, such as violaxanthin and antheraxanthin, in cancer therapy. These carotenoids interfere with resistance mechanisms such as those involving ABC transporters and destroy the structural integrity of cancer cell membranes, showing a dual mechanism. Their antiproliferative effect was strongly enhanced by the combination with epirubicin, especially in cell models of lymphoma and leukemia. These carotenoids act mainly as sensitizers, increasing the effectiveness of chemotherapeutics interfering with the defense mechanisms of cancer cells, in contrast to direct lethal effects of fucoxanthin.

Despite the promising data on the synergistic potential of carotenoids with chemotherapeutic agents, there is a notable lack of in vivo studies demonstrating these benefits. In vivo research is essential to gain a comprehensive understanding of how these combinations work in complex biological systems, taking into account factors such as drug metabolism, bioavailability, and potential systemic toxicity. Such studies are crucial for bridging the gap between encouraging in vitro findings and clinical applications and providing critical insights into the therapeutic efficacy and safety profiles of these compounds under physiological conditions.

### 4.3. Recent Advances in the Development of Alternative Novel Therapies

To enhance the effectiveness of natural compounds and overcome the drawbacks of traditionally used therapies for malignant diseases, as well as improve tumor imaging techniques, researchers have been investigating alternative approaches [134]. On the one hand, various attempts have been made to enhance the effectiveness of microalgae for tumor treatment due to their unique photosynthetic abilities and adaptability. On the other hand, the tumor microenvironment is characterized by acidic pH, tumor hypoxia, and vascular abnormalities, often causing tumor resistance to commonly used therapies [135].

For example, *Chlorella vulgaris* was coated with cell membranes derived from engineered dendritic cells. In this way, their functionality was improved by combining their ability to produce oxygen in situ within hypoxic tumor environments and reduce hypoxia with the immunostimulatory properties of the dendritic cell membranes [31]. This concept was further pursued in a more recent study where “photosynthetic micron robot” was produced by combining *C. vulgaris* with genetically engineered dendritic cell membranes overexpressing TNF ligand proteins (OX40L, 4-1BBL, and CD70) [29]. The hybrid algae-radiation-laser combination therapy system effectively suppressed tumor growth by blocking the HIF1α/VEGF axis to inhibit angiogenesis and proliferation and induce apoptosis. In another study, *C. vulgaris* was used as an oxygenator to relieve tumor hypoxia in situ through oxygen generation [136]. The surface of the microalga was engineered with calcium phosphate (CaP) to develop an efficient delivery system (CV@CaP) that activated the oxygenation reaction in tumors.

Several studies have also been conducted to manipulate algal gene expression resulting in an increased yield of the desired metabolites [30]. Diatom *Thalassiosira pseudonana* was genetically engineered to produce the IgG-binding domain of protein G on the biosilica surface, which allowed the selective targeting of neuroblastoma and B-lymphoma cells and reduced tumor growth in a mouse xenograft model. The results suggest that genetically modified diatom-derived biosilica could be a promising platform for delivering anticancer drugs to specific tumor sites [28].

Moreover, researchers have been investigating the potential of photosensitizer-based therapies to reduce immune-suppressive conditions caused by tumor cells [134]. Photosensitizers, such as chlorophyll derived from microalgae, could be used for bioimaging and photodynamic treatment (PDT) [137]. For example, the surface of *Spirulina platensis* was modified to develop a biohybrid system for tumor targeting and imaging due to its high chlorophyll content and ability to generate ROS upon 650 nm irradiation [135]. Overall, PDT works best for solid tumors because targeted light application can activate the photosensitizers that produce ROS and cause tumor cell death [138,139]. In contrast, the diffuse distribution of malignant cells in hematological malignancies, such as leukemia and lymphoma, poses a major challenge for effective light delivery and the application of photosynthetic pigments [140]. To our knowledge, only the study of Oka et al. [141] suggested that 5-Aminolevulinic acid (5-ALA), a precursor for the synthesis of chlorophyll, can exert selective cytotoxic activity against ATL cells, with minor influence on normal lymphocytes. However, recent investigations suggest that encasing photosensitizers derived from natural sources within a shell of nanoparticles could advance their efficiency. Besides more specific delivery of photosensitizers to malignant cells due to the presence of nanocarriers, these nanoparticles can overcome tumor hypoxia and enhance PDT efficiency [134].

### 4.4. Algae-Assisted Nanoparticles

To overcome drug resistance and improve the treatment of aggressive or metastatic diseases, increasingly sophisticated drug delivery systems with advanced targeting capabilities are being developed. Algae-derived nanoparticles stand out in this field due to their simplicity, efficiency, and environmentally friendly production methods while overcoming challenges such as drug resistance and off-target toxicity [142]. Although algae have the ability to absorb certain metal ions or micro- or nanoparticles, algal nanoparticles are usually synthesized via “green synthesis” using algal extracts as natural reducing and stabilizing agents (Figure 3). Algal phytochemicals, including hydroxyl, carboxyl, and amino groups, play a crucial role in the synthesis of inorganic nanoparticles. As bio-reducing agents, these functional groups facilitate the reduction in metal ions while improving the stability and biocompatibility of the resulting nanomaterials [143].

One example of algae-assisted synthesis is the development of hyaluronan/zinc oxide (HA/ZnO) nanocomposites. These are hybrid materials consisting of zinc oxide (ZnO) nanoparticles in combination with hyaluronan, a naturally occurring sugar molecule that is often used in biomedical applications due to its compatibility with human tissue. In this case, the synthesis of ZnO nanoparticles is supported by extracts of the brown macroalga *Sargassum muticum*, which are used to produce zinc oxide nanoparticles. These hybrid nanocomposites show selective cytotoxicity against HL-60 leukemia cells with an IC_50_ of 6.25 μg/mL after 72 h. By activating caspases-3 and -7, the HA/ZnO nanoparticles efficiently induce apoptosis while arresting the cell cycle in G2/M phase. The stability and functionality of these nanoparticles are enhanced by the presence of hydroxyl and carboxyl groups from algal extracts. These chemical groups play a crucial role in ensuring the structural integrity and reactivity of the nanoparticles. Despite their ability to kill leukemia cells, the HA/ZnO nanoparticles show no toxicity to normal human lung fibroblast cells (MRC-5), highlighting their potential for targeted leukemia therapy [142].

Another example of algae-assisted nanoparticle synthesis is the production of gold nanoparticles (AuNPs) using water extracts from the macroalga *Sargassum glaucescens*. In this synthesis, the amino and carboxyl groups contained in the algal extracts act as natural stabilizers that ensure their biocompatibility. With an IC_50_ of 10.32 μg/mL, these nanoparticles show strong, selective cytotoxicity against CEM-SS leukemia cells. Their effect is based on the intrinsic apoptotic pathway, which is characterized by the activation of caspases 3 and 9. The amino and carboxyl groups present in the algal extract are essential for the stabilization of these nanoparticles and ensuring their biocompatibility. Importantly, the gold nanoparticles did not exhibit cytotoxicity towards normal mammary epithelial cells (MCF-10A), highlighting their safety and potential for further applications [144].

Magnetic iron oxide nanoparticles (MNPs) synthesized with *Sargassum muticum* extracts are another promising therapeutic candidate. These nanoparticles, which consist of magnetite (Fe_3_O_4_), are formed by the reduction of iron (III) ions by biomolecules in the algal extract, such as hydroxyl and amino groups. Fe_3_O_4_ MNPs show significant cytotoxicity on Jurkat cells with an IC_50_ of 6.4 μg/mL. Flow cytometry shows that Fe_3_O_4_-MNPs effectively induce apoptosis and arrest the cell cycle in sub-G1 phase. The time-dependent activation of caspase-3 and caspase-9 confirms their apoptotic mechanism [143].

While previous studies focused on macroalgae for nanoparticle synthesis, research by Delalat et al. [28] highlights the potential of diatoms as valuable organisms for drug delivery. Diatoms have a unique porous silica shell known as biosilica, which provides a large surface area and high mechanical stability, making them suitable carriers for therapeutic agents. Biosilica from *Thalassiosira pseudonana* has been genetically engineered to bind antibodies and transport chemotherapeutics, enabling selective delivery to cancer cells. Despite the difficulties posed by the systematic and dispersed nature of these diseases, diatom-based carriers present an intriguing opportunity for the development of novel therapeutic approaches. Specific chemical groups on the biosilica surface, such as hydroxyl and carboxyl groups, increase the ability to bind drugs. When loaded with SN38 micelles (a chemotherapeutic agent) and labeled with antibodies targeting cancer cells, the artificial biosilica showed excellent results. In a xenograft model with mice, a single dose of antibody-functionalized biosilica significantly reduced the tumor by up to 53%, demonstrating its efficacy in targeted drug delivery [28]. In conclusion, the biodistribution and stability of biosilica, as well as the possibility of immune responses or reactions with non-target organs, remain significant obstacles that limit its therapeutic application and require further research.

In another study [145], the functionalization of biosilica from *Thalassiosira pseudonana* was also demonstrated but, in this case, with sodium alendronate, a bisphosphonate widely used in the clinical treatment of bone metabolic disorders such as osteoporosis, fibrous dysplasia, myeloma, and bone metastases. The high osteoconductivity of this functionalized biosilicon dioxide promoted osteoblast activity while preventing osteoclast-mediated bone resorption. These materials minimize systemic side effects and improve local therapeutic efficacy by utilizing the porous structure and surface functional groups of biosilica to create a controlled drug release mechanism. These results, which focus primarily on bone regeneration, demonstrate that it is possible to modify diatom biosilica for targeted drug delivery in hematological malignancies.

The pH-dependent and sustained release of silica nanoparticles (SiNPs) loaded with chemotherapeutic drugs (such as doxorubicin) aligns with the principles of microalgae-based platforms previously described. For instance, the hybrid “photosynthetic micron robot” system, which leverages the hypoxic tumor environment, could incorporate SiNPs engineered for pH-sensitive drug release [146]. This approach would optimize the therapeutic efficacy of the drug by ensuring its release under acidic conditions prevalent in tumors, thereby minimizing off-target effects. Similarly, engineered biosilica from diatoms, like *Thalassiosira pseudonana*, could serve as a carrier for doxorubicin, offering sustained and localized drug delivery while targeting specific tumor cells.

### 4.5. In Vivo Studies

While in vitro studies have demonstrated the anticancer and immunomodulatory properties of algal extracts or individual compounds, the translation of these findings to in vivo models remains limited. In vivo research is needed to establish the therapeutic potential of compounds derived from algae to verify their efficacy and ensure their safety in biological systems.

For example, Yim et al. [147] isolated a high-sulfate containing exopolysaccharide, p-KG03, from the red microalga *Gyrodinium impudicum* (strain KG03) and examined its immunostimulatory effects using in vitro and in vivo assays. In addition, the activities of natural killer cells obtained from the treated mice against mouse lymphoma cells (YAC-1) were investigated. Mice received a single dose of p-KG03 (100 or 200 mg/kg body weight), followed by the isolation of peritoneal macrophages at 10 or 20 days after the treatment. In response to the increase in natural killer cells and macrophage production levels of IL-6, IL-1β, and TNF-α, the in vivo treatment with p-KG03 mediated the nonspecific immune functions in a dose-dependent manner. Furthermore, these results indicate that p-KG03 has immunostimulatory effects and enhances the antitumor activities of natural killer cells and macrophages.

Fucoidan, another polysaccharide isolated from brown macroalga *C. okamuranus*, induced an immunomodulatory effect in 8-week-old female BALB/c mice [148]. In this research, mice were continuously treated with different fucoidan concentrations (102.5–1025.0 mg/kg) for six weeks. During the experiment, macrophage phagocytosis, cytokine production, immune cell proliferation, and serum antibody concentration were measured. The results showed that fucoidan increased macrophage activity and also the activity of immune cell in the spleen. Moreover, an increase in the cytokines IFN-γ and IL-2 and the serum antibodies IgA, IgG, and IgM levels were observed, suggesting that fucoidan can modulate humoral and cellular immunity.

Macroalga *C. okamuranus* was also used for the extraction of fucoxanthin, an algal carotenoid. In addition to in vitro studies, Yamamoto et al. [71] demonstrated that treatment with fucoxanthin suppresses tumor growth in vivo and reduces effusions in body cavity.

Despite the promising in vitro results on the anticancer potential of algal extracts and compounds, in vivo studies investigating their efficacy remain extremely limited. This lack of research using animal models to explore the potential of various algae-derived substances in the treatment of hematological malignancies underscores the urgent need for further investigation. Future animal studies need to confirm in vitro data and ensure their relevance for therapeutic applications. Such integration is crucial to establish the safety, efficacy, and potential clinical applicability of these bioactive compounds.

### 4.6. Clinical Studies

Algal compounds, especially peptides, have been used directly (modified and unmodified) or conjugated with specific antibodies in cancer therapies [6]. In targeted cancer treatment, antibody–drug conjugates enable the delivery of cytotoxic agents to specific antigen-expressing tumor cells through monoclonal antibodies [149].

So far, several compounds found in algae have also been tested in different clinical trials. An overview of those trials is presented in Table 5.

Among those compounds is kahalalide F, a depsipeptide derived from macroalga *Bryopsis* sp., currently in Phase III of a clinical trial for treating prostate, lung, and skin cancer [155]. In addition, curacin A isolated from *Lyngbya majuscula* and dolastatin 10 isolated from *L. majuscula* and *Symploca hynoides* have been evaluated in clinical studies for the cancer treatment or used as basic structures for the synthesis of more efficient analogs/derivatives [156].

Curaxins, compounds related to curacin A, have been tested in several clinical studies as antitumor agents. Specifically, curaxin CBL0137 has been evaluated in a Phase I trial in participants with unresectable advanced or metastatic or solid malignancies (study ID: NCT01905228) [157]. Also, CBL0137 in combination with ipilimumab and nivolumab has been examined for therapeutic application in melanoma patients, but the study is currently paused for safety cohort review (study ID: NCT05498792) [158].

CBL01377 was also planned to be evaluated in a trial targeting participants with previously treated lymphoma, but, unfortunately, this clinical study was terminated (study ID: NCT02931110) [159]. Currently, the only active study is a Phase I/II trial aiming to test CBL01377 for the treatment of refractory or relapsed solid tumors. The study is in the recruiting status, with estimated completion in late 2026 (study ID: NCT04870944) [160].

In the field of hematological malignancies, one of the first FDA-approved treatments for classical HL and anaplastic large-cell lymphoma is brentuximab vedotin (BV) [161,162]. BV is an antibody–drug conjugate comprising a CD30-targeted antibody, linked to the microtubule disruptor monomethyl auristatin E (MMAE, also known as a dolastatin 10 derivate) via protease cleavable linker. CD30 is one of the factor receptors for tumor necrosis, typically expressed in specific hematological malignancies and primarily manifested on the surface of various malignant cells, e.g., Reed–Sternberg cells in HL, different T-cell cancers, systematic anaplastic large cell lymphoma cells, embryonal carcinomas, and other hematological malignancies [161,163].

MMAE is a synthetic analog of dolastatin 10, a natural pentapeptide consisting of valine, dolavaline, dolaproline, dolaisoleucine, and C-terminal amine dolaphenin. In comparison, MMAE has the same peptide backbone in its structure as dolastatine 10 but a different C-terminus (dolaphenine is substituted with (1S,2R)-(+)-norephedrine) [78,164]. MMAE binds to tubulin, which leads to the disruption of the microtubule network and consequently triggers cell cycle arrest in the G2/M phase and apoptosis [78,165].

Another antibody–drug conjugate consisting of a monoclonal antibody against a B cell receptor CD79b and MMAE is polatuzimab vedotin (PV) [166]. PV, in combination with rituximab and bendamustine, is FDA approved for the treatment of refractory or relapsed large B-cell lymphoma [78].

Monomethyl auristatin F (MMAF) is another synthetic dolastatin 10 derivate with a phenylalanine instead of dolaphenine at its C-terminus. MMAF is a part of belantamab mafodotin, a BCMA-targeted therapy approved by the FDA for the treatment of triple-class refractory MM patients [78,126].

Overall, some of the major bottlenecks in the development and implementation of novel therapies for hematological malignancies that could be tested in clinical studies are safety concerns, limited access to patient, and complex regulatory requirements [126,167]. Moreover, patients with hematological malignancies are often repeatedly exposed or refractory to different treatments such as proteasome inhibitors, CD38 monoclonal antibodies, and immunomodulatory drugs. This has led to the investigation and development of more targeted approaches.

## 5. Conclusions and Future Perspectives

The review highlights the promising therapeutic potential of algae-derived bioactive compounds in the treatment of hematological malignancies while addressing key challenges that hinder their translational applicability. A major obstacle to the extrapolation of results into clinical practice is the predominance of findings derived from in vitro research, with limited transition to in vivo studies and clinical trials. In addition, many in vitro studies evaluate algal extracts without performing detailed chemical analyses to pinpoint the specific compounds responsible for the observed bioactivities.

To fully harness the potential of algae in hematological malignancy treatment, it is crucial to implement the systematic screening and identification of bioactive compounds, followed by rigorous validation through in vivo and clinical studies. Equally essential is the systematic evaluation of long-term toxicity, safety, and off-target effects through preclinical and clinical studies focusing on bioavailability and pharmacokinetics.

Furthermore, the synergistic potential of algae-derived bioactive molecules in combination with existing therapies for hematological malignancies remains underexplored. Investigating such combinations could potentially enhance therapeutic efficacy, reduce drug dosage, attenuate drug resistance, and decrease relapse rates.

In summary, overcoming these limitations could unlock the vast untapped potential of microalgae and macroalgae as sources of bioactive compounds, paving the way for transformative advancements in the treatment of hematological malignancies. Such progress holds the promise of introducing safer, more innovative, and highly effective therapeutic options.

## Figures and Tables

**Figure 1 cancers-17-00318-f001:**
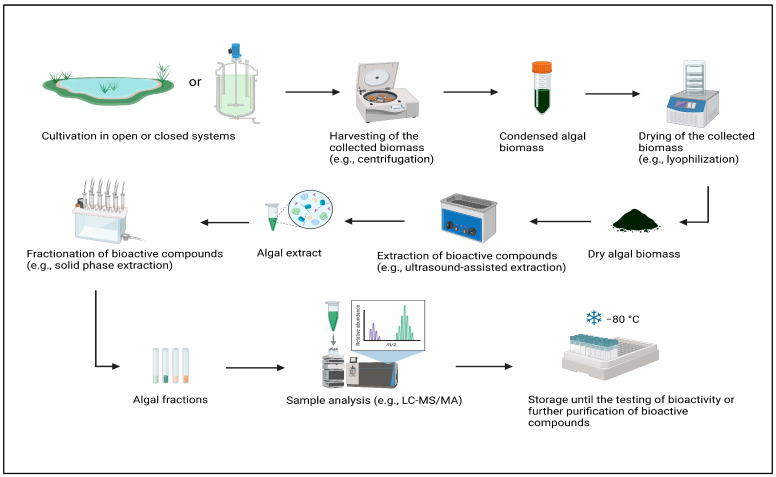
An overview of general steps to be taken before determining the bioactive properties of compounds present in collected algal biomass.

**Figure 2 cancers-17-00318-f002:**
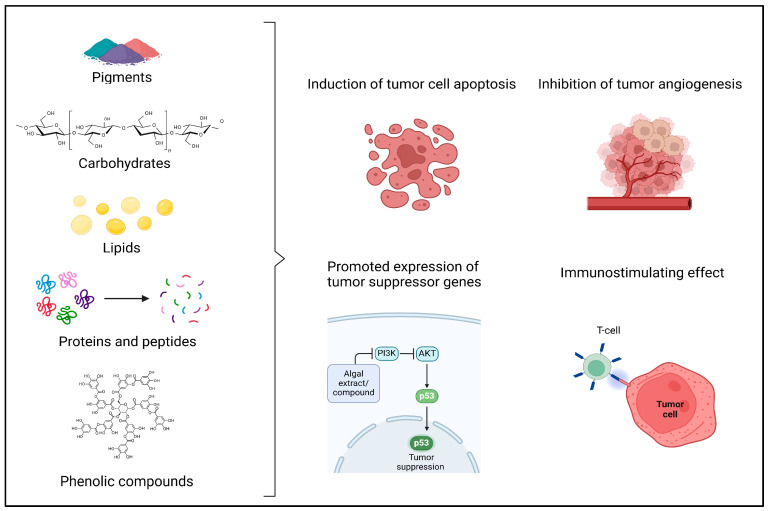
Algal compounds with the most promising anticancer activities and their effects on tumor cells.

**Figure 3 cancers-17-00318-f003:**
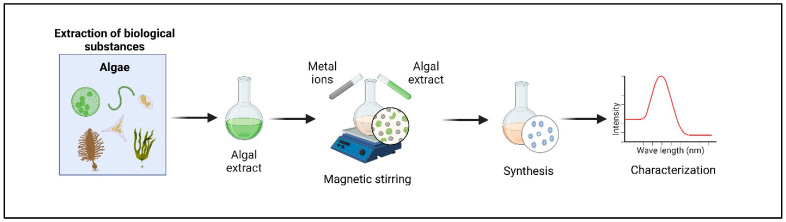
Green synthesis of algal nanoparticles.

**Table 1 cancers-17-00318-t001:** An overview of algal extracts that could potentially be used in treating hematological malignancies.

	Algal Species	Tested Extract	Cell Line	IC_50_ Value	Reference
Macroalgae	*Bifurcaria bifurcata*	CE	Jurkat, Daudi, K562	N/A, tested conc. 100 μg/mL	[58]
	*Caulerpa lentillifera*	EM	HL-60	315.50 μg/mL	[49]
	*Caulerpa lentillifera*	EE	HL-60	1085.01 μg/mL	[49]
	*Caulerpa lentillifera*	EAM	HL-60	2505.90 μg/mL	[49]
	*Caulerpa lentillifera*	EAE	HL-60	3003.00 μg/mL	[49]
	*Caulerpa lentillifera*	HM	HL-60	3805.50 μg/mL	[49]
	*Caulerpa lentillifera*	HE	HL-60	3003.00 μg/mL	[49]
	*Colpomenia sinuosa*	EAE	U937, HL-60	63.45 and 53.35 μg/mL	[57]
	*Cystoseira tamariscifolia*	CE	Jurkat, Daudi, K562	N/A, tested conc. 100 μg/mL	[58]
	*Desmarestia ligulata*	CE	Jurkat, Daudi, K562	N/A, tested conc. 100 μg/mL	[58]
	*Dictyota dichotoma*	CE	Jurkat, Daudi, K562	N/A, tested conc. 100 μg/mL	[58]
	*Dictyota dichotoma*	AE	K562	9.76 ± 0.34 µg/mL	[59]
	*Galaxaura oblongata*	EAE	U937, HL-60	83.79 and 132.73 μg/mL	[57]
	*Halidrys siliquosa*	CE	Jurkat, Daudi, K562	N/A, tested conc. 100 μg/mL	[58]
	*Halimeda discoidae*	EAE	U937, HL-60	139.37 and 226.35 μg/mL	[57]
Microalgae	*Chlorella sorokiniana*	ME	L5178Y-R	460.0 ± 21.5 μg/mL	[60]
	*Chlorella* sp.	HE	K562	21.37 ± 2.98 μg/mL	[52]
	*Microcystis aeruginosa*	ME	HL-60, K562	N/A, tested conc. 1, 5, 10, 50, and 500 μg/mL	[53]
	*Nannochloropsis oculata*	ME	HL-60, K562	N/A, tested conc. 1, 5, 10, 50, and 500 μg/mL	[53]
	*Phaeodactylum tricornutum*	ME	HL-60, K562	N/A, tested conc. 1, 5, 10, 50, and 500 μg/mL	[53]
	*Scenedesmus* sp.	ME	L5178Y-R	362.9 ± 13.5 μg/mL	[60]
	*Skeletonema marinoi*	ME/DMSO	K562	N/A, tested conc. 0.5 mg/mL and 0.75 mg/mL	[54]
	*Stichococcus bacillaris*	ME	HL-60, K562	N/A, tested conc. 1, 5, 10, 50, and 500 μg/mL	[53]

AE—acetone extract; CE—crude extract; EAE—ethyl acetate extract; EAM—ethyl acetate macerate; EE—ethanol extract; EM—ethanol macerate; HE—hexane extract; HM—hexane macerate; ME—methanolic extract; ME/DMSO—methanolic extract solubilized in DMSO. N/A—information not available.

**Table 2 cancers-17-00318-t002:** Antitumor properties of macroalgal pigments tested in vitro using cell lines related to hematological malignancies.

Algal Species	Compound	Cell Line	IC_50_ Value	Reference
*Cladosiphon okamuranus*	Fucoxanthin	PEL	N/A, tested conc. 10 μM	[71]
*Cladosiphon okamuranus*	Fucoxanthinol	PEL	N/A, tested conc. 5 μM	[71]
*Codium fragile*	Siphonaxanthin	HL-60	N/A, tested conc. 5, 10, and 20 μg/mL	[73]
*Ishige okamurae*	Fucoxanthin	HL-60	12.1 μM	[72]
*Solieria filiformis*	R-phycoerythrin	HL-60	61.27 μg/mL for LPE and 130.0 μg/mL for WPE	[76]
*Undaria pinnatifida*	Fucoxanthin	HTLV-1-infected T-cells	3.31 ± 1.44 μM	[70]
*Undaria pinnatifida*	Fucoxanthinol	HTLV-1-infected T-cells	1.24 ± 0.43 μM	[70]

LPE—R-phycoerythrin derived from lyophilized seaweed extract; WPE—R-phycoerythrin derived from wet seaweed extract.

**Table 3 cancers-17-00318-t003:** In vitro studies of algal polysaccharides with potential anticancer activity.

	Algal Species	Compound	Cell Line	IC_50_ Value	Reference
Macroalgae	*Laminaria ochroleuca*	Sulfated polysaccharides	U937	3.72 mg/mL	[91]
	*Porphyridium cruentum*	Sulfated polysaccharides	U937 and HL-60	1676.74 g/mL and 1089.63 g/mL	[100]
	*Sargassum polycystum*	Fucoidan	HL-60	84.63 ± 0.08 g/mL	[37]
Microalgae	*Tetraselmis suecica*	EPS derived from autotrophic (A) and heterotrophic (B) microalgal culture	HL-60	36 μg/mL (A) and 68 μg/mL (B) for acidic EPS and 1784 μg/mL (A) and 5183 μg/mL (B) for total EPS	[97]
	*Gymnodinium* sp.	GA3P	K562	0.017 μg/mL for GA3Pl+ and 0.015 mg/mL for GA3Pl−	[99]
	*Isochrysis galbana*	(1 → 3, 1 → 6)-β-D-glucan	U937	N/A, tested conc. 50 and 100 μg/mL	[103]

**Table 4 cancers-17-00318-t004:** An overview of other specific algal compounds that could potentially be used for the treatment of hematologic malignancies.

	Algal Species	Compound(s)	Cell Line	IC_50_ Value	Reference
Macroalgae	*Bryopsis* sp.	Kahalalides P2 and Q3	HL-60	N/A	[67]
	*Kappaphycus alvarezii*	Lectin Siye	HL-60	3.949 μg/mL	[121]
	*Laminaria japonica*	Phlorotannins	P388	120 μg/mL	[128]
	*Perithalia capillaris*	bis-prenylated quinone	HL-60	0.34 μM	[129]
	*Tricleocarpa jejuensis*	(±)-(E)-12-hydroxyoctadec-10-enoic acid (HOEA)	U937	47.96 μg/mL	[109]
Microalgae	*Crypthecodinium cohnii*	DHA	HL-60	74 µM	[108]
	*Lyngbya majuscula*	Hantupeptin A	MOLT-4	32 μM	[131]
	*Lyngbya majuscula*	Somocystinamide A	Jurkat and CEM	3 nM and 14 nM	[114]
	*Lyngbya majuscula*	Lagunamides A, B, and C	P388	6.4, 20.5, and 24.4 nM	[119,120]
	*Nostoc* sp.	Criptophycin 1	L1210	N/A, tested conc. 5 and 10 μg/mL	[118]
	*Phaeodactylum tricornutum*	Nonyl 8-acetoxy-6-methyloctanoate	HL-60	65.15 µM	[110]

**Table 5 cancers-17-00318-t005:** Clinical studies testing commonly found compounds in algae (source: ClinicalTrials.gov).

Tested Compound(s)	Targeted Diseases or Patients	Study ID (nct Number)	Phase	Status	Trial Type	Reference
Fucoidan	Stage III/IV head and neck squamous cell carcinoma	NCT04597476	II	Recruiting	Supportive Care	[150]
Fucoidan	Cancer survivors	NCT06295588	N/A	Not yet recruiting	Treatment	[151]
Lectin	Patients with solid tumors after failure of standard therapy	NCT00006477	Phase I	Completed	Treatment	[152]
Omega-3	Monoclonal gammopathy of undetermined significance and smoldering multiple myeloma	NCT05640843	N/A	Recruiting	Prevention	[153]
Phycocyanin	Metastatic gastric cancer	NCT05025826	N/A	Recruiting	Prevention	[154]

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
