# Peer review of "Unlocking the Therapeutic Potential of Algae-Derived Compounds in Hematological Malignancies"

_cancers, 2025, doi:10.3390/cancers17020318_

Round 1

Reviewer 1 Report

Comments and Suggestions for Authors

This is a very comprehensive review of the potential of algae-derived compounds in the therapy for the commonest hematological malignancies. It is well written, provides good insights with a very thorough but somewhat lengthy explanation of the background of algae. The manuscript can benefit with perhaps an abbreviated and condensed background of the biology but expansion into the clinical studies (limitations, future use), with more clinical relevance to suggest future application of algae derived compounds for hematological malignancies as suggested by the title, this would make it more meaningful. Graphical abstracts and tables provided are clear.

Author Response

REVIEWER #1

This is a very comprehensive review of the potential of algae-derived compounds in the therapy for the commonest hematological malignancies. It is well written, provides good insights with a very thorough but somewhat lengthy explanation of the background of algae. The manuscript can benefit with perhaps an abbreviated and condensed background of the biology but expansion into the clinical studies (limitations, future use), with more clinical relevance to suggest future application of algae derived compounds for hematological malignancies as suggested by the title, this would make it more meaningful. Graphical abstracts and tables provided are clear.

We thank the reviewer for the positive comments and thoughtful suggestions.

Regarding the background on the biology of algae, we appreciate the feedback and acknowledge that it may appear somewhat lengthy. However, we believe that further condensing this section could compromise the clarity and comprehensiveness of the manuscript, which aims to provide a strong foundation for understanding the therapeutic potential of algae-derived compounds.

In response to the reviewer’s suggestion, we have expanded the discussion on clinical studies, focusing on their limitations and future applications, as suggested by the title of the review. This content is now presented in greater detail in a newly added section titled “4.3. Recent Advances in the Development of Alternative Novel Therapies” (lines 838–844 and 848–866).

Reviewer 2 Report

Comments and Suggestions for Authors

The authors discussed the antitumor properties of algae, focusing on their potential in treating hematologic malignancies. The topic is interesting.

Photosynthetic pigments such as chlorophyll, which are rich in algae, can be used as natural photosensitizers. Photosensitizers can generate significant levels of reactive oxygen species (ROS), which possess the ability to induce cell death in tumor cells. These facts need to be discussed.

Algae can be genetically engineered to enhance their biological activity and modify their physical and chemical properties. Although the authors highlighted this point, they did not discuss much in detail. More literature should be cited and it should be discussed in detail.

Microalgae strongly adsorb certain metal ions or micro- or nanoparticles, and the surface can be easily modified for targeted delivery of drugs. This fact should be discussed in detail.

Since diatoms are highly porous, they provide multiple adsorption sites for chemotherapeutic agents and other nanoparticles, making them useful for cargo delivery. The text should cover diatoms in detail.

In vivo delivery in complex environments by using diatoms as novel silica carriers may also face difficulty. The authors should discuss these aspects.

Can the pH-dependent and sustained release of SiNPs loaded with adriamycin be beneficial?

Suppression of metastatic signaling pathways should be discussed in detail.

Can the low cost of microalgae be beneficial?

Photodynamic therapy is adversely affected by tumor hypoxia. Therefore, improving the hypoxic microenvironment of tumors would greatly enhance the effectiveness of photodynamic cancer therapy. The authors should discuss this aspect.

Microalgae can be applied to bioimaging due to their easy culture and good biocompatibility. This new aspect needs to be discussed in detail.

Author Response

REVIEWER #2

The authors discussed the antitumor properties of algae, focusing on their potential in treating hematologic malignancies. The topic is interesting.

  1. Photosynthetic pigments such as chlorophyll, which are rich in algae, can be used as natural photosensitizers. Photosensitizers can generate significant levels of reactive oxygen species (ROS), which possess the ability to induce cell death in tumor cells. These facts need to be discussed.

Response: Thank you for the suggestion. Although photosynthetic pigments like chlorophyll and their role as photosensitizers are interesting, the systemic and dispersed nature of hematological malignancies limits their use. For solid tumors, photodynamic treatment works best because targeted light application can activate photosensitizers, that produce reactive oxygen species (ROS) and cause tumor cell death. In contrast, the diffuse distribution of malignant cells in hematological malignancies, such as leukemia and lymphoma, poses a major challenge for effective light delivery and the application of photosynthetic pigments. Consequently, the number of studies investigating the application of algae or its metabolites as natural photosensitizers is very limited. We have included a paragraph (lines 654-666) in the revised manuscript to address this topic.

  1. Algae can be genetically engineered to enhance their biological activity and modify their physical and chemical properties. Although the authors highlighted this point, they did not discuss much in detail. More literature should be cited and it should be discussed in detail.

Response: Thank you for the suggestion. We have included a paragraph (lines 640-659) in the revised manuscript to address the role of genetic engineering and other types of modifications in enhancing the biological activity of algae and modifying their physical and chemical properties. We have expanded the discussion by incorporating additional relevant literature that highlights recent advancements in the manipulation of algae.

  1. Microalgae strongly adsorb certain metal ions or micro- or nanoparticles, and the surface can be easily modified for targeted delivery of drugs. This fact should be discussed in detail.

Response: We thank the reviewer for the suggestion. We have already devoted a section to the discussion of algae-assisted nanoparticles. However, the text was additionally modified following the recommendation (Lines: 644-647, 657-659, 667-672, 678-681).

  1. Since diatoms are highly porous, they provide multiple adsorption sites for chemotherapeutic agents and other nanoparticles, making them useful for cargo delivery. The text should cover diatoms in detail. In vivo delivery in complex environments by using diatoms as novel silica carriers may also face difficulty. The authors should discuss these aspects.

Response: We appreciate the reviewer’s suggestion. We have already mentioned the cargo delivery potential of diatoms in paragraph 4.4. (Lines: 721-728). Following the reviewer’s advise, we highlighted the challenges of using silica within an in vivo system (lines: 730-737).

  1. Can the pH-dependent and sustained release of SiNPs loaded with adriamycin be beneficial?

Response: The pH-dependent and sustained release of SiNPs loaded with chemotherapy drugs (such as doxorubicin or adriamycin) can be beneficial by enabling selective drug release in the acidic tumor microenvironment, minimizing off-target effects and systemic toxicity. Within the study, we discussed the possibility of incorporating pH-sensitive SiNPs into hybrid "photosynthetic micron robots" to optimize therapeutic efficacy. Biosilica derived from genetically engineered diatoms can serve as a biocompatible carrier for localized and sustained drug delivery, enhancing tumor targeting and treatment outcomes. The text was modified in accordance (lines: 744-752).

  1. Suppression of metastatic signaling pathways should be discussed in detail.

Response: We thank the reviewer for the suggestion. The disscussion on the supression of the metastatic pathways is added throughout the manuscript (lines 151-155, 307-310, 580-583).

  1. Can the low cost of microalgae be beneficial?

Response: Yes, and this is mentioned in Chapter 2, the low cost of microalgae offers significant advantages, especially in the context of therapeutic applications. Additional sentence has been added to this section to emphasize this point (lines 71-72). Microalgae are not only inexpensive to cultivate, but also environmentally friendly, as their production does not compete with agricultural resources. Their rapid growth rate makes large-scale production possible. In addition, microalgae can be cultivated under various conditions, e.g. in wastewater or salt water, which reduces production costs and negative environmental impact. They are rich in bioactive compounds such as antioxidants, vitamins, and essential fatty acids, which contribute to their therapeutic potential.

  1. Photodynamic therapy is adversely affected by tumor hypoxia. Therefore, improving the hypoxic microenvironment of tumors would greatly enhance the effectiveness of photodynamic cancer therapy. The authors should discuss this aspect.

Response: We thank the reviewer for the suggestion. We have included a paragraph (lines 634-637 and 643-646) in the revised manuscript to address the impact of tumor hypoxia on the efficiency of photodynamic therapy.

  1. Microalgae can be applied to bioimaging due to their easy culture and good biocompatibility. This new aspect needs to be discussed in detail.

Response: We appreciate the reviewer’s suggestion. We agree that bioimaging presents a new aspect worth exploring further. However, existing studies referring to the application of algae to bioimaging in treating any malignancy, not just hematological, are very scarce and insufficient for an elaborate discussion. Therefore, we briefly discussed the use of microalgae for bioimaging and photodynamic treatment in Chapter 4.3 (lines 655-658).

Reviewer 3 Report

Comments and Suggestions for Authors

The manuscript entitled ‘Unlocking the Therapeutic Potential of Algae-Derived Compounds in Hematological Malignancies' is very interesting. The text is well-written. The references need to appear throughout the text in this configuration [1]. Change in all text. And the reference list is incorrect with the journal format. Please, correct.

I recommend the manuscript for acceptance with minor revisions.

Line 159: ‘FDA’. Is the first time that this acronym appears on the text. Please, provide its significance.

Line 315 and Table 2: ‘Cladosiphon okamuranus Tokida’. The name Tokida is the author that described this specie. If you put for this specie you need to put for all species cited throughout the text. I suggest delete this Tokida.

Line 527: mycosporine-like amino acids are not a phenolic group. Change this. You can create another subsection to explain about MAAs.

Line 780: ‘onceptualization’. Please, correct to ‘Conceptualization’.

Author Response

REVIEWER #3

The manuscript entitled ‘Unlocking the Therapeutic Potential of Algae-Derived Compounds in Hematological Malignancies' is very interesting. The text is well-written. The references need to appear throughout the text in this configuration [1]. Change in all text. And the reference list is incorrect with the journal format. Please, correct. I recommend the manuscript for acceptance with minor revisions.

  1. Line 159: ‘FDA’. Is the first time that this acronym appears on the text. Please, provide its significance.

Response: Thank you for the comment. „Food and Drug Administration“ has been added (lines 58 and 59).

  1. Line 315 and Table 2: ‘Cladosiphon okamuranus Tokida’. The name Tokida is the author that described this specie. If you put for this specie you need to put for all species cited throughout the text. I suggest delete this Tokida.

Response: Thank you for the comment. „Tokida“ has been removed.

  1. Line 527: mycosporine-like amino acids are not a phenolic group. Change this. You can create another subsection to explain about MAAs.

Response: Thank you for the suggestions, we apologize for the oversight. Since mycosporine-like amino acids (MAAs) do not fall into any of the compound categories discussed in this review, and there is a lack of studies related to hematological malignancies involving MAAs, we have decided to omit the discussion of MAAs from the paper.

  1. Line 780: ‘onceptualization’. Please, correct to ‘Conceptualization’.

Response: Thank you for the comment. Corrected.

Round 2

Reviewer 2 Report

Comments and Suggestions for Authors

The authors revised the paper according to the comments.